# Proposal for Mathematical and Parallel Computing Modeling as a Decision Support System for Actuarial Sciences

Marcos dos Santos [1,2], Carlos Francisco Simões Gomes [3], Enderson Luiz Pereira Júnior [3], Miguel Ângelo Lellis Moreira [2,3,*], Igor Pinheiro de Araújo Costa [2,3] and Luiz Paulo Fávero [4]

1   Systems and Computing Department, Military Institute of Engineering, Rio de Janeiro 22290-270, Brazil
2   Operational Research Department, Naval Systems Analysis Centre, Rio de Janeiro 20091-000, Brazil
3   Production Engineering Department, Federal Fluminense University, Rio de Janeiro 24210-240, Brazil
4   School of Economics, Business, and Accounting, University of São Paulo, São Paulo 05508-010, Brazil
*   Correspondence: miguellellis@hotmail.com

**Abstract:** This paper aims to find the actuarial tables that best represent the occurrences of mortality and disability in the Brazilian Armed Forces, thus providing a better dimensioning of the costs of military pensions to be paid by the pension system. To achieve this goal, an optimization software was developed that tests 53 actuarial tables for the death of valid military personnel, 21 boards for entry into the disability of assets, and 21 boards for mortality of invalids. The software performs 199 distinct adherence tests for each table analyzed through linear aggravations and de-escalations in the probabilities of death and disability. The statistical–mathematical method used was the chi-square adherence test in which the selected table is the one with the null hypothesis "observed data" equal to the "expected data" with the highest degree of accuracy. It is expected to bring a significant contribution to society, as a model of greater accuracy reduces the risk of a large difference between the projected cost and the cost observed on the date of the year, thus contributing to the maintenance of public governance. Additionally, the unprecedented and dual nature of the methodology presented here stands out. As a practical contribution, we emphasize that the results presented streamline the calculation of actuarial projections, reducing by more than 90% the processing times of calculations referring to actuarial projections of retirees from the armed forces. As a limitation of the study, we emphasize that, although possibly replicable, the database was restricted only to the Brazilian Armed Forces.

**Keywords:** operational research; actuarial science; parallel computing; computer science; decision-making process

## 1. Introduction

The decision-making process in political environments involves different areas, interconnecting strategic, tactical, and operational levels in favor of a direction aligned with the objectives in a given problematic situation [1–9]. The high-level decision-making sphere is complex, where the given form of solution can generate influences not only in the political sphere but also impacts other areas of society [10,11]. In this context, the decision analysis in complex environments, integrating multiple stakeholders analyzing aspects relevant to the problem is common [12,13], enabling, from multiple perspectives, a consensus in the process [14,15].

With the involvement of multiple scenarios and circumstances, the increase in complexity in a given analysis becomes noticeable [16,17], with different points of view as to the importance or influence of a decision variable, although, it is necessary to consider it in favor of a substantial evaluation and greater assertiveness in the final decision [18–20].

It is usual to apply operational research (OR) techniques and methods, to provide the optimization of the resource environment [21], enabling the cost reduction and, consequently, a better understanding, analysis, and solution of complex problems [22,23]. In

this context, the decision concerning public pension is one of the most important economic decisions [24], if we consider that it defines, at an aggregate level, the performance of the economy investments [25,26].

In this scenario, the social security schemes are generally very relevant to the well-being of large numbers of beneficiaries [27]. Thus, one sought to create an unprecedented methodology capable of providing the best possible accuracy in the projection of revenues and costs of the military pension system over a 75-year time horizon in the context of actuarial science.

This analysis is restricted to the military pensions defined by Law No. 3765, dated May 4, 1960, of Brazil. With regard to social security, it emphasizes that, besides the legal and social aspect, there is also the economic aspect [28]. It cannot be neglected that rights generate costs, especially the rights that demand a "doing" of the public administration [29,30].

It should be noted that the military of the armed forces are not part of the social security system, published in Brazil, since in the military pension system, there is no accumulated equity to bear the future costs, and it is therefore financed by regime without capital accumulation [31,32]. Corroborating this statement, it points out that, legally, the federal military does not have a social security system, and its proceeds are fully funded by the National Treasury [33]. Hence the importance of keeping track of the projected values and values effectively contributed by the Union, year by year [34].

Thus, extensive research was conducted in the social security system (SSS) and in the general social security system (GSSS) [35]. It was verified that the methodology, parameters, and biometric assumptions adopted by these regimes could not be used for the social protection of military personnel (SPMP), in view of the peculiarities of the military career [36,37].

The armed forces have a considerable mass of financial and biometric data of their pensioners that needs to be submitted to its own actuarial mathematical model [38] to obtain the result of the actuarial projection of 75 years, which is a legal requirement that must be met annually [39]. In this context, the objective of the present study is to reduce the computational times necessary to obtain the results of the referred projection, through the application of operations research, parallel computing, and software engineering techniques.

This paper is divided into six sections. The first presents a brief introduction on the problem. The second describes the problem and the proposed methodology. The third is the implementation of the proposed methodology. The fourth presents the results of the proposed heuristic called the adhesion cube. The fifth brings some details of the armed forces actuarial software (AFAS) and the sixth makes some final considerations with their respective conclusions.

## 2. Background

There is the regular compulsory contribution of active and inactive military, whose rate is 7.5% in the social security system [40]. The projection of the collection and the constitutional cost with proceeds aims to confer transparency and predictability on the obligations of the armed forces, in order to guide the formulation of policies to maintain the long-term fiscal sustainability [41].

The theme of social security became important for Brazilian society since the year 1995 (the year of the establishment of the real as the Brazilian currency). The deficit has billions of shares that grow systematically year by year [42]. Analyzing a historical series, since 1995, this deficit was around one billion reais in 1997, and in 2017 it surpassed two hundred and sixty billion reais (USD 250,000.00) according to the Secretariat of Social Security of the Ministry of Finance [42]. The parameters for the calculation of the pension deficit in the referred historical series remained unchanged, and such numbers were confirmed and validated by the Federal Court of Accounts (TCU) [43]. In order to maintain the solvency of the Brazilian State, the Federal Government presented a series of proposals aimed at reducing the medium to long-term welfare deficit, or at least interrupting its growth trend [44].

To conduct the application of actuarial science to the costs of pensioners of the armed forces, it is necessary to study and design the constitutional costs of the current assets, inactive, and pensioners [45]. However, it is known that demographic–actuarial projections cannot be absolutely accurate. However, long-term projections are carried out by the Brazilian government and constitute a set of decisions of strategic relevance for the country; its limits have to be more clearly stated and the use of more recent projection and scenario construction techniques should be the basis of a more robust and reliable decision support system [46].

Indemnity pensions or special pensions of the group of ex-combatants (soldiers who fought in wars) and political amnesties (after democratization in 1985) are not included in this study, because if they were considered, they would cause unnecessary distortions, given the specificity of each one. The work is restricted to the military pensions defined by Law No. 3765. The proposed model was initially tested with data from the Strategic and Management Information Bank (SMIB) for the month of October/2015 and the actuarial projections were made for the period from 2017 to 2091. The article proposes a methodology capable of evaluating, with the best possible accuracy, the income and costs of pensioners of the armed forces over a 75-year horizon, based on existing data, using actuarial science and operational research [47].

### 2.1. Actuarial Tables

Actuarial tables are tables that include the social and biometric characteristics of a given sample for analysis of expectations and risk in actuarial science. The tables selected in an actuarial study must effectively represent the biometric events (such as death, disability, and illness) that affect the analyzed population. They should be chosen based on the historical experience and perspectives of the sample. Most studies of seasonal variation in mortality rely on aggregated counts of deaths at the population level. The use of actuarial tables that are out of step with reality can result in cumulative actuarial losses or gains over time, generating structural imbalances in the system [48].

According to [49], the actuarial table is the basic tool for analyzing population evolution, representing the oldest demographic model in use, historically used to measure the longevity of a population [50]. The earliest known tables date back to the 3rd century, but modern actuarial ones were developed from the 17th century [51]. Since then, its practical applications were diversified, with new relationships developed and functions improved [52]. The actuarial tables are the foundation for any product in the social security area and may come from data from a demographic censuses or the insurance companies' experience [53].

### 2.2. Goodness-of-Fit (GoF) Tests

GoF tests are statistical methods that allow you to examine how well a data sample matches a given distribution [54,55]. The purpose of these tests is to assess whether two frequency distributions are approximately identical or whether they can be considered heterogeneous [56]. It is considered as a statistical test of probability distribution model, in which the observed proportions are adjusted to the expected proportions [57], mathematically deduced, or established according to some [58].

GoF tests consist of verifying the suitability of a probabilistic model to a data set. In the GoF tests, there is a null hypothesis $H_0$ that X, a random variable, follows a declared probability law $F(x)$. The techniques of these tests consist of mathematical models to measure the conformity of the data of a sample, that is, set of values of x with the hypothetical distribution; or, equivalently, with its discrepancy about it [59]. In other words, the basic concept is that, given a random sample of size n, observed from a random variable X, it is desired to test the null hypothesis $H_0$ that the sample follows a certain distribution function $F(x)$, confronting it with the alternative hypothesis $H_1$, that the sample does not follow the distribution function $F(x)$:

**$H_0$.** *X has distribution F(x)* vs. **$H_1$.** *X has no distribution F(x).*

In the formal structure of the adherence test, the null hypothesis $H_0$ can be a simple hypothesis when F(x) is specified completely or $H_0$ can provide an incomplete specification and then it will be a compound hypothesis.

Among the most used GoF tests in actuarial studies and the social security market, the chi-square test was applied in this paper [48].

### 2.3. Chi-Square Adherence Test

The performance of the adherence tests has the objective of evaluating if two frequency distributions are approximately identical or if they are considered heterogeneous [60]. The test consists of analyzing whether the distribution of decrements (death and disability) given by the analyzed actuarial tables represent or not the distribution of decreases in the databases of the Brazilian Armed Forces, that is, Brazilian Navy (BN), Brazilian Army (BA), and Brazilian Air Force (BAF).

The objective of the test is to compare the differences between the (E) and the observed (O) frequencies, considering as observed the deaths, or inflows, occurring in a specific time horizon for each age group, inactive and pensioners [61].

The estimation of average death or disability is performed by multiplying the probabilities associated with each age, according to the tables, by the number of individuals exposed to the risk of this same population [62].

Approaches are useful to improve accuracy [63]. In order to test whether the calculated discrepancies are statistically significant, the $x^2$ compares with the same factor ($x^2_{critical}$) obtained from the chi-square distribution table [64]. In order to obtain the results, a 5% significance level was adopted, that is, a 5% probability of rejection of the null hypothesis, which considers that the observed frequency is equal to the mean frequency [65], that is, the distribution of deaths or entry into invalidity of the Brazilian Armed Forces identical to the distribution of a certain actuarial table. According to [66], the index, $x^2$ is calculated by the formula expressed in (1).

$$X^2 = \sum_{i=1}^{n} \frac{(O_i - E_i)^2}{E_i} \tag{1}$$

where:

- $X^2$ = chi-square test statistics;
- $n$ = maximum age in the actuarial tables;
- $O_i$ = observed frequency of deaths/disability with age $i$;
- $E$ = average death/disability frequency with age $i$.

The lower the divergence between the observed frequency and the mean frequency, the lower the statistic $X^2$ and the probability of not rejecting the hypothesis of adherence between the actual mortality/disability experience and the board adopted as the premise is greater. After calculating the $X^2$, the critical $X^2$, is verified, taking into account the level of significance adopted and the degrees of freedom considered in the test. In the study, each age group represents an independent observation of the sample. Thus, the number of degrees of freedom of the statistic $X^2$ is represented by the number of age ranges used subtracted from one, due to the intrinsic characteristic of the test model used.

### 3. Methodology

Considering the Brazilian political–economic scenario, it became mandatory to generate the results of actuarial projections as quickly as possible in order to support the decision of the armed forces high command [67]. Thus, the research in hand proposes the methodology explained in Figure 1.

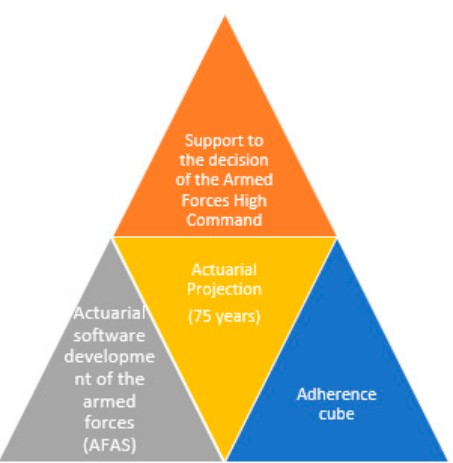

**Figure 1.** Methodology for the actuarial projection of the costs with pensioners of the armed forces.

In the central part of Figure 1, the armed forces have the need to carry out the actuarial projection of the pensioners by determination of the TCU. This projection must have the best accuracy possible, which motivated the creation of the conceptual model of the adherence cube. With the computational effort imposed by the adherence cube, it was necessary to use parallel computing in order that the projections were calculated quickly [68].

In addition, it was necessary to create an application that allowed the entry of the data of the armed forces and generated the results of the projections. This application was called the actuarial software of the armed forces (AFAS). This technical–methodological procedure, formed by the adherence cube, the use of parallel computing, and the development of AFAS [69], aims to support the high command of the armed forces through precise results generated in time to support the decision.

Several researchers already developed and published studies related to the estimation of actuarial tables using adhesion tests, such as chi-square, mean square deviation and Kolmogorov–Smirnov, such as [70] and [71]. However, no technical studies were found on how to optimize the estimation of the actuarial tables to be adopted, considering increases and reductions in the probabilities of death and disability [72], which is widely accepted by current legislation. The following steps were followed (methodological proposal):

- Step 1 creating a trusted database;
- Step 2 define/choose a heuristic to be applied to the data;
- Step 3 database heuristic adherence test;
- Step 4 selection of the best model for the study;
- Step 5 selection of the computational model; and
- Step 6 analysis of results.

### 3.1. Creation of SMIB

In the development of the remuneration studies, the MD realized that the lack of detailed information on military payroll expenses was the major hindrance to negotiations with the Ministry of Planning, Budget and Management (MPBM) and, consequently, the major problem for the monitoring and evaluation of the financial impact of the proposed adjustments or of changes in the legislation in force. In order to reduce this deficiency and make it possible to monitor the effects of provisional measure (PM) n° 2.215/2001 [73], SMIB was created.

From the year of its creation, SMIB was perfected and managed by the Secretariat of Co-ordination and Institutional Organization (SCIO), through the advisory office, Department of Coordination, Organization and Legislation (DCOL) and, currently, the Compensation Division, so that it serves the MD as a managerial tool in the development of several studies.

With this, SMIB is the only database that allows consolidated calculations of the three forces, providing the calculation of financial impacts arising from measures that may alter

the remuneration policy of the military and its pensioners. It is possible to understand the strategic importance of SMIB to the armed forces.

The analysis of large amounts of data by man is not feasible without the aid of appropriate computational tools. Therefore, it is essential to provide tools that help the man in the task of analyzing, interpreting, and relating these data, so that strategies can be developed and selected for action in each field of application [74].

Initially, the platform used for the SMIB was SQL Server 2000 from the Microsoft Corporation, which contain data as of January 2001, taking up space of approximately 20 gigabytes, 25% of the total dedicated server capacity (80 gigabytes). New data are stored per month, with approximately 200 megabytes, or 0.25% of the total server space, which is equivalent to the annual data addition of 3% of the storage capacity of that server. The information available in the SMIB also allows the MD to attend the various administrative queries requested and the institutional demands required of other government or private bodies, as shown in Figure 2.

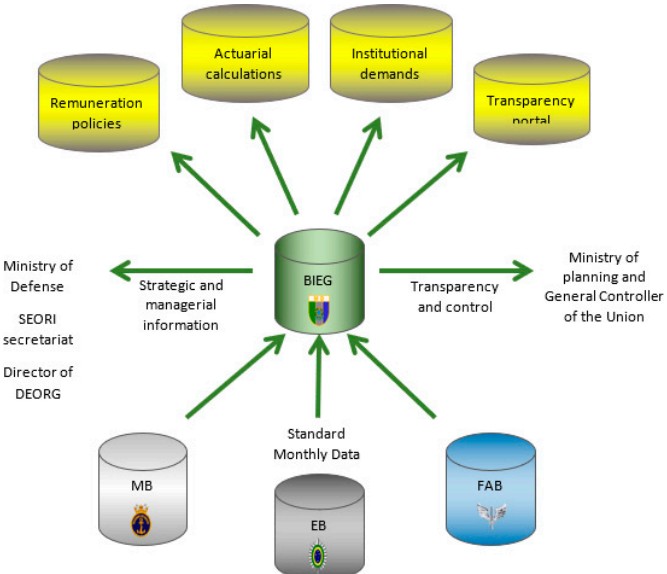

**Figure 2.** Strategic and Management Information Bank (SMIB).

In order to improve the SMIB, some minimum requirements were established, mainly to improve the quality of management information contained in the bank, not forgetting the reliability and standardization of the data. The requirements are:

- Greater consistency of information based on the maintenance of a unique historical basis;
- Higher reliability of the database, these being correct and applicable in any situation;
- Ability to consolidate information, with the possibility of aggregating remunerative installments of various forms;
- Ability to detail the remuneration structure of the military, with the possibility of explaining the composition of the military remuneration; and
- Ability to detail payroll by nature of expenses (NE), which provides greater precision in comparisons with the federal government's integrated financial administration system (FGIFAS).

In this sense, in order to keep the database reliable, meetings are held annually to discuss possible improprieties in some data sent by the forces and to discuss suggestions for improvements and improvements to correct these inconsistencies. With each modification introduced, new instructions are given to provide data, aiming at maintaining the standardization of information. Before sending the data to the MD, the forces are able to verify the results obtained by these routines and, when necessary, rectify them.

The criticisms of the database were established based on general rules and that the occurrences resulting from the application of them are in fact inconsistencies, since there may be specific cases in each force that justify some occurrences without representing impropriety.

### 3.2. Heuristics

#### 3.2.1. The Probabilistic Multidecrements

The pension and social security systems of countries that require the solvency of social security entities require the measurement of contributions and future payments of active, inactive, and pensioners, whose technical nomenclature is the actuarial projection of revenues and expenses.

In order to comply with rigid regulations established by [75], in this area, the manager needs to determine the mortality and disability tables that best represent the expectations of decrements, that is, death and disability of the study population.

The graph shown in Figure 3, shows that an active individual is subject to joint probabilities of dying, or being invalidated. Each vertex of the graph establishes a state in which the military can meet (active, reserved, married, retired, with children, etc.). It is often said that this individual is subjected to probabilistic multidecrements.

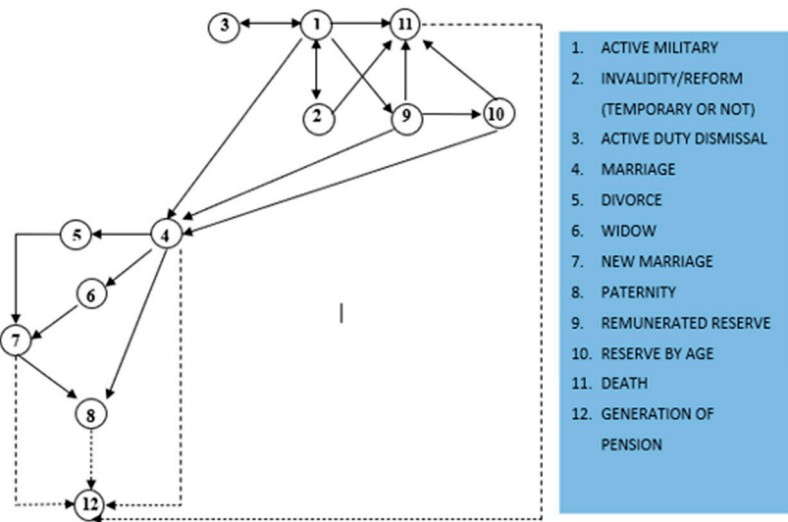

**Figure 3.** Biometric and social states.

Thus, Figure 3 shows that:

- An active military (state 1) can be declared invalid (state 2), marry (state 4), go to reserve (state 9), resign (state 3), or die (state 11);
- An invalid military (state 2) can return to active (state 1), marry (state 2), or pass away (state 11);
- If the military member resigns (state 3), he leaves the system and does not generate a pension;
- A married soldier (state 4) can divorce (state 5), become a widower (state 6), or become a parent (state 8). Once married, the dotted line shows that the military can generate a pension (state 12);
- A divorced military (state 5) may enter into a new marriage (state 7);
- A widowed soldier (state 6) may contract a new marriage (state 7);
- A military who contracted a new marriage (state 7) can become a parent state 8);
- A military man who became a father (state 8) can generate a pension (state 12) as indicated by the dotted line;
- A military in the Remunerated Reserve (state 9) can contract marriage (state 4), go to the reserve by age (state 10), or die (state 11);
- A military in the reserve by age (state 10) may marry (state 4) or die (state 11); and

- ■ Generation of pension (state 12) can be generated by marriage (state 4), by a new marriage (state 7), and by paternity (state 8) once the military passed away (state 12).

### 3.2.2. The Actuarial Tables

De Witt in the Netherlands, and Grauns and Halley in England, advanced in the studies of probability and demographics related to human longevity. According to [76], only in 1815, Milne created the first table of life that contemplated probabilities of death ($q_x$) and survival ($p_x$).

Such a table was drawn up on the basis of the observed mortality of the inhabitants of Carlisle, England. According to [77], numerous tables were drawn up for different regions and countries around the world because of their crucial importance for the analysis of sociological, population, economic, and biometric factors.

For [78], every product, whether in the social security area or the living area, is supported by the mortality tables. Starting from a closed number of participants, called the "root", in which the gender can be taken into account, the mortality table reveals the number of people living annually at each age, that is, it is a board determined by annual mortality or survival rates. The construction of these tables can be derived from the insurers' experience or using the census data.

Currently, professionals in the area have access to dozens of mortality and disability tables for different populations and different breeding dates. In this way, the professional must choose among the numerous existing tables that present the greatest accuracy and precision in relation to the decreases observed in the population of the private pension entity under study [79].

The actuarial tables help in the performance of various sociological and economic calculations [80], such as calculating the life expectancy of assets, inactive and invalid, the average number of people that will be invalidated, population density, measurement of economically active population, and others. The most important actuarial tables for social security entities are: the mortality table of invalid, table of disability, and table of mortality of invalids, which are detailed below [81].

### 3.2.3. The Mortality Tables

According to [82], the actuarial calculus works by providing means to calculate insurance premiums related to life and social security costs [83]. The most traditional element of this technique is the mortality table, whose function is to give life expectancy for a given (whole) age in the interval [0.1]. This probability distribution is called the survival function, represented in (2):

$$S(x) = P[X > x] \tag{2}$$

For [84], in the last decades, actuaries and demographers used increasingly sophisticated methods for predicting mortality.

Mortality tables, also known as survival tables, are intended to represent the actual probability of death ($q_x$) of a given population, and such death probabilities are determined by the age of the individuals in the population.

The mortality table or life table is the most complete tool for analyzing the mortality of a population. Another important aspect of the study of mortality is the analysis of the various socioeconomic characteristics, such as the composition of the workforce, age population, and the regulation of social security systems [85].

### 3.2.4. Invalidity Entry Tables

Invalidity entry tables are intended to represent the actual probability that a valid ($i_x$) is due to a fortuitous event. These probabilities are defined according to the age of the individuals in the population. In addition to the probabilities of invalidity, these tables provide the average number of people who theoretically will be invalidated by age in a particular closed population, in addition to other statistical information that can be extracted by algebraic calculations [86].

### 3.2.5. Tables of Invalidity Mortality

The mortality tables for invalids are applied exclusively to the participants who are no longer active because they contracted some incapacity that does not allow them to exercise their work activity. Such boards are intended to represent the actual probability of death of this specific group of participants by age [87]. In addition to the probabilities of death of invalids, these tables provide the average number of deceased and survivors by age in a given closed population, the expected life expectancy of an incapable participant, as well as additional information that can be obtained through math operations [88,89].

### 3.2.6. Selection of the Model

The methodology consists of the application of 199 adhesion tests for each actuarial board analyzed, that is, the adhesion of the board, and 198 variations of the board are verified, which represents 99 aggravations and 99 different reliefs. The worsening rates vary from 1% to 99%, which increase the chances of death/disability of all ages from 1% ($[q_x \text{ or } i_x] \times 0.99$), respectively.

Thus, the reductions tested consist of reducing the odds of all ages from 1% ($[q_x \text{ or } i_x] \times 0.99$) to 99% ($[q_x \text{ or } i_x] \times 1.01$). The actuarial tables of mortality and invalidity that were analyzed in this study are presented in Tables 1 and 2. The specifics of the actuarial tables used in this research can be found on the Society of Actuaries (SOA).

**Table 1.** Valid mortality tables used.

| CS0-41 | CSO-58 | CSO-80 | AT-49 | AT-50 | AT-55 |
|---|---|---|---|---|---|
| AT-71 | American Experience | GAM-1971 | SGB-51 | SGB-71 | SGB-75 |
| IAPC | Hunter Semitropical | Rentiers Français | Grupal Americana | USTP-61 | GKM-70 |
| GKM-80 | ALLG-72 | X-17 | CSG-60 | Prudential 1950 | GAM 1994 Male |
| RP-2000-1992 Base-Male Aggregate | AT-2000 | AT-2000 F | AT-83 | AT-83 male | UP-84 |
| UP94Men | UP94Woman | UP-94 MT-M-ANB | GRM-80 | GRF-80 | GRM-95 |
| GRF-95 | BR-EMSsb-v.2010-m | BR-EMSsb-v.2010-f | BR-EMSmt-v.2010-m | BR-EMSmt-v.2010-f | BR-EMSsb-v.2015-m |
| BR-EMSsb-2015-f | BR-EMSmt-2015-m | BR-EMSmt-2015-f | CSO2001MALE | CSO2001FEMALE | IBGE-2011-M |
| IGBE-2011-F | IBGE-2011 | IBGE-2012-M | IBGE-2012-F | IBGE-2012 | - |

**Table 2.** Invalidity and death tables for invalids used.

| Invalidity and Mortality Tables for Invalids Used | | | | | |
|---|---|---|---|---|---|
| IAPB-57 Weak | IAPB-57 Strong | Zimmermann | Zimmermann (Ferr. Germans) | Zimmermann (Empre. Write.) | Grupal Americana |
| Álvaro Comings | TASA-1927 | Prudential (Ferr. Retired.) | IBA (Railways) | Muller | Hunter's |
| IAPB-57 (AJUST/ITAU) | Winklevoss | Bentzien | IAPC | IAPB-57 | ALLG72 |
| USTP61 | Rentiers Français | X17 | - | - | - |

In this way, tests are performed for three different groups, valid mortality, invalidity mortality, and invalidity, in order to determine the actuarial tables that have the best adherence to data coming from a specific database of the Ministry of Defense (MD).

Initially, the probabilities referring to the original mortality/disability tables, as shown in Tables 1 and 2, are structured in matrices, through a computational application developed in C, for each group studied. In Brazil, the category of military professionals does not resemble any other professional category because it has its own characteristics, especially regarding biometric data. Such specificities require the development of said application.

Table 1 presents the main existing actuarial tables [53], which were analyzed in order to obtain the one most adherent to the data of the Brazilian military.

These matrices have two dimensions, the first one referring to the tables being analyzed (47 for mortality of valid and 21 for invalidity and mortality of invalids) and the second corresponding to the possible ages of the participants (0–125 years).

The matrix shown in Figure 4 is considered the base matrix to achieve the 99 aggravated boards and the 99 additional mitigated boards, which, when generated by multiplicative terms, result in a three-dimensional matrix.

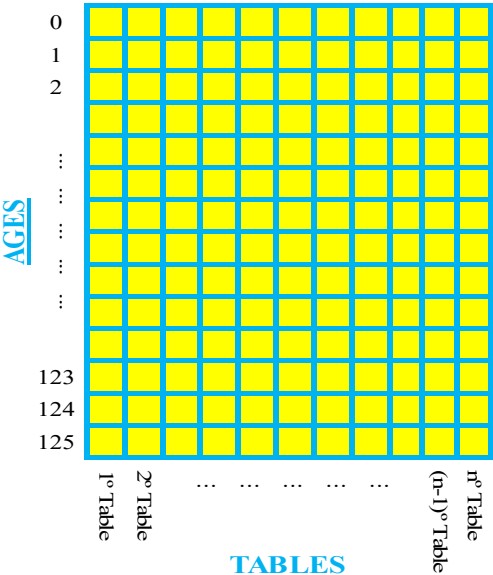

**Figure 4.** Matrix of original actuarial boards.

Thus, the cube shown in Figure 5 is a conceptual model called the "Adherence Cube", represented by a three-dimensional matrix that contains the original, aggravated, and undeserved death/invalidity probabilities.

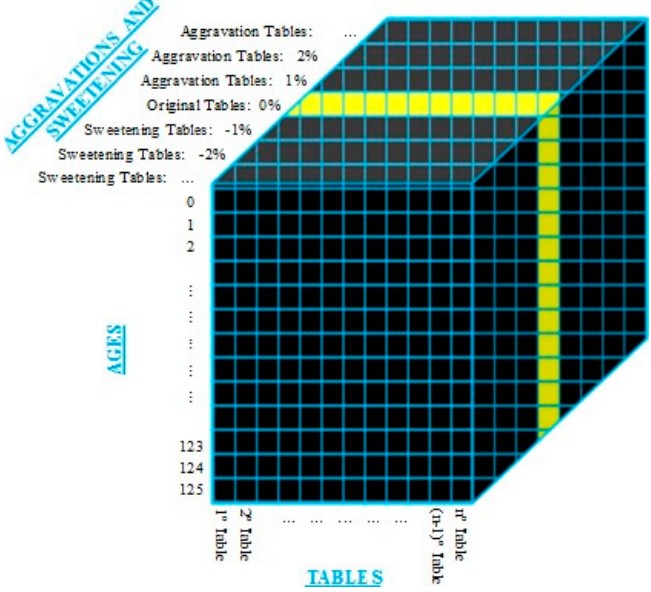

**Figure 5.** Adherence Cube.

Therefore, the first dimension represents the actuarial tables presented in Tables 1 and 2, the second dimension means the ages of the participants, and the third dimension corresponds to the original table with its respective aggravations and redemptions.

After determining all odds of death/disability, the application starts performing the adhesion tests according to (1). A grip test is applied for each actuarial table generated, considering that it varies from age zero to age 125 years. In this sense, 9353 tests ([47 different actuarial tables] × [1 original board + 99 aggravated boards + 99 allowable boards]) are necessary to perform the validity test for adherence of valid mortality tables. Accordingly, for conducting the invalidity and invalidity mortality tests, there are 4179 tests ([21 boards] × [1 original board + 99 aggravated boards + 99 boards granted]).

In this way, the board that has the best adherence to the data is the $x^2$, calculated as less than the $x^2$ (true null hypothesis), and it has the lowest statistical value among all the results obtained. Therefore, it is concluded that the distribution of deaths or disability entry of the Brazilian Armed Forces is more adherent to the distribution of a certain actuarial table.

The developed actuarial application provides as a result of the actuarial calculations a two-dimensional matrix, signaling in green all the tables in which the null hypothesis was considered true; however, the board with the best adherence to the AF mortality/disability events is indicated in red and exemplified in Figure 6.

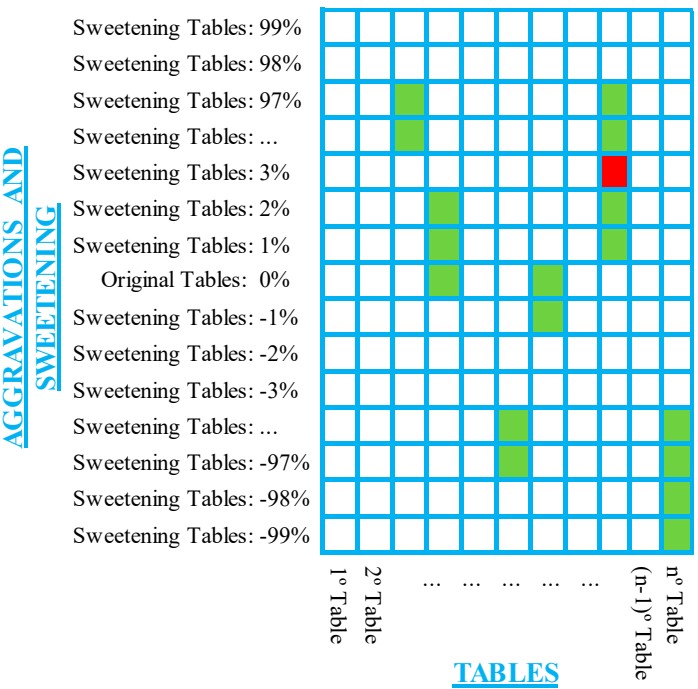

**Figure 6.** Results obtained with the developed application.

### 3.3. Computational Model

The actuarial calculus is a mathematical method that uses financial, economic, and probability concepts to measure the number of resources and contributions necessary to pay future benefits of the insureds of the social security funds and institutes, also called social security own regimes, which seeks the balance between the financial results and the actuarial projection. The maintenance of this balance is of paramount importance, especially in times of economic crisis.

The actuarial projections of income and expenses for a pension and pension entity are intended to quantify the estimated future costs of payment of benefits and the estimated future income from participant contributions. These projections are important for entities and governments to provision such monetary amounts for subsequent years, ensuring

actuarial balance and reducing the risk of illiquidity. As exposed in [90], when contributions are properly defined, planning tends to have a financial–actuarial equilibrium [45].

Pension and social security institutions project future costs and revenues over a 75-year time horizon; however, nothing prevents the projection from being calculated at 80 or 100 years. The main assumptions that impact this actuarial calculation are: mortality table of invalids, table of mortality of invalids, table of disability, turnover rate, wage growth, rate of inflation, and replacement of assets.

### 3.4. Parallel Computing

Faced with the limitations of sequential computing, we opted for parallel computing, so that the results of the actuarial projections of the armed forces could be generated in an acceptable time, in order to support the senior administration of the armed forces in matters related to pension reform with the government federal.

In order to support the development and use of a computational application for the actuarial calculation, we sought knowledge in areas related to parallel computing, focusing on the creation of a computational environment capable of propitiating the actuarial projections in a timely manner. According to [91], parallel computing is a kind of computing in which multiple calculations are performed in several parallel, rather than sequential, processing units [92,93]. The idea here is to replace a large mass of processing in a single CPU by smaller processing in several CPUs [94].

The adoption of a methodology that uses parallel computing resources in the development of an application to calculate actuarial projections can, by extrapolating the results achieved within the scope of the armed forces, considerably shorten the time needed to obtain results and reduce the costs of organizations with the contracting of third-party services.

The exact percentage of time reduction achieved by parallel computing in actuarial problems will depend on a variety of factors, including the size of the problem, the number of processors used, and the efficiency of the algorithms used. However, substantial reductions in processing time can often be achieved, sometimes on the order of several orders of magnitude.

Although we have not found specific studies on reducing processing times in actuarial science, the academic literature presents some studies that use parallel computing to reduce processing time in complex problems, such as:

Cader et al. [95] focused on software-based acceleration for plagiarism detection using CPU/GPU. The authors gained 45× speed-up compared to the CPU. Hossain and Assiri [96] proposed a facial expression recognition framework with the incorporation of parallelism, and the processing time speeds up three times faster. López et al. [97] proposed the use of RGB point clouds estimated from structure from motion (SfM) as the input for building thermal point clouds. The authors reported up to 96.73% less processing time. Morishima [98] proposed a subgraph-based anomaly detection method to perform the detection using a part of the blockchain data. According to the author, the proposed method was 11.1× faster than an existing GPU-based method without lowering the detection accuracy. Zhang [99] presented a method that parallelizes Bayesian computation using distributed computing on Apache Spark across a cluster of computers. According to the author, the distributed algorithm achieved as much as 65 times performance gain over the non-parallel method.

These papers demonstrate the potential benefits of parallel computing in reducing processing time in actuarial problems, although the exact percentage of time reduction will vary depending on the specific problem and implementation. Analyzing these findings, we obtained a good starting point for understanding the use of parallel computing in reducing processing time in actuarial problems and the potential for speed-up. The exact percentage of reduction in processing time may vary depending on the specific problem and the parallel computing method used.

### 3.4.1. Parallel Programming with C #

Current architectures may contain two, four, six, or more complex processing cores, and are very much employed in exploring parallelism at the instruction and thread level. In theory, such architectures could be extended to tens or even hundreds of cores in the future, but they run into two main obstacles: the difference between processor and memory speeds, known as memory walls, and excessive heat generation due to high frequencies necessary for the operation of such architectures [100].

Multi-core processors (color) were on the market for many years and are currently available on most devices. However, many developers continue to do what the same thing: create programs that use a single stream of isolated sequential control inside a program called a thread. This means that not all processing power available on the machine is used. Most currently commercialized computers have a four-core and four-core processor, or six-core and eight-core processors. By purchasing a computer with such features, users pay for extra processing power. However, by providing a program that uses a single thread, the developer does not allow the use of this extra processing capacity.

Kirk and Hwu [101] point out that it is a simple task to achieve a ten-fold speed up when an application makes use of data parallelism. A computation is said to be parallel when a program runs on a multiprocessor machine in which all processors share access to available memory, i.e., the same address on different processors corresponds to the same memory location. Golov and Rönnbäck [102] make similar comments.

Multi-threaded processing is not new to experienced C # developers, but it is not always easy to develop programs that use all the available processing power. In addition, the evolution of programming languages facilitated the work of developers by simplifying the implementation of parallel programming.

In order to advance the use of parallel programming, it is important to establish two important concepts: synchronous execution and asynchronous execution. A good insight into these two modes of execution is basic knowledge to improve the performance of your applications. When executing a synchronous operation, the program executes all the tasks in sequence, as shown in Figure 7. When firing the execution, each task will only be executed after the previous task ends.

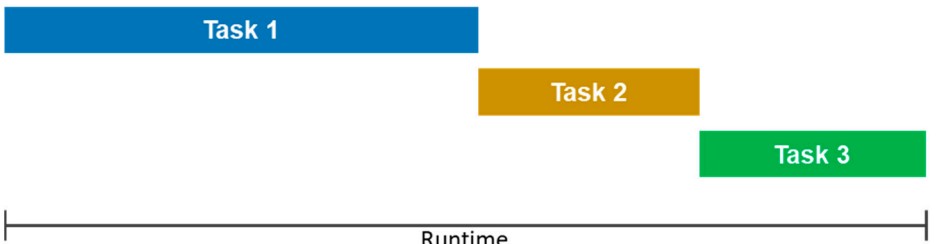

**Figure 7.** Synchronous execution.

When executing an operation in asynchronous mode, the program triggers the tasks when necessary, and they are started and closed concomitantly to the execution of other tasks, as shown in Figure 8.

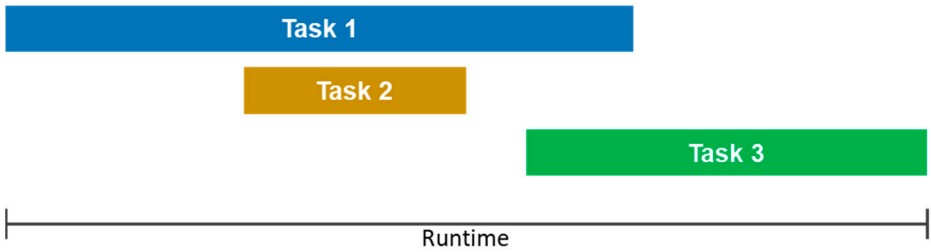

**Figure 8.** Synchronous execution.

As it induces image analysis, the same tasks running asynchronously will require a shorter execution time than when executed in a synchronous manner, for the simple fact that there is no need for one task to "wait" for another task to start. Waiting, in this case, would only be acceptable if there were a dependency relationship between the tasks; that is, if a task depended, for example, on a calculation that another task, still in execution, is producing.

From the previous paragraph it is possible to deduce that the use or not of the parallel programming should be evaluated by the developer. In some situations, its use can be quite beneficial by decreasing the execution times and producing the results. However, in other situations, its unbalanced use may even degrade application performance. Since the implementation of parallel programming with C # requires the explicit insertion in the code of specific instructions of parallels, it is up to the developer to decide when and where to use such instructions.

The question may arise as to why many developers still choose to run synchronously if asynchronous execution takes less time. The answer to such a question is not simple. What can be said straight away is that with asynchronous programming, the developer has some new challenges:

Synchronize tasks. Assuming that in Figure 7 it is necessary to start a task only after the other two are finished, it would be necessary, in this case, to create a wait mechanism to wait for all tasks to finish before performing the new task.

Solve competition problems. If a shared resource exists, such as a list that is written to a task and read into another task, you would need to create a mechanism to ensure that the list is kept in a known state.

Adapt to a new programming logic. Since there is no logical sequence, tasks can end at any time and you no longer have control of which ends first.

Asynchronous programming requires a paradigm shift on the part of the developer. Its adoption, however, has some advantages. One of the most significant is the non-crash of the user interface (UI), as the tasks can be executed in the background. Another advantage is the ability to use all the cores of the machine, making better use of its resources.

### 3.4.2. Asynchro Programming with Async and Await

It is possible to avoid performance bottlenecks and improve the overall response of software using asynchronous programming. However, traditional techniques for writing asynchronous applications can be tricky, making it difficult to develop, debug, and maintain.

Asynchrony is essential for activities exposed to a potential block, such as when the application accesses the web. Access to a web resource is sometimes slow or subject to delays. If such activity is blocked within a synchronous process, the entire application will be on hold. In an asynchronous process, the application can continue to perform another task, which does not rely on the web resource, until the task exposed to a potential block ends.

Async and await, keywords in C #, are the core of asynchronous programming. Using these two words, one can use the features of the .NET framework or the Windows runtime to create an asynchronous method almost as easily as creating a synchronous method.

Figure 9 shows part of an asynchronous routine. When using the async keyword, you can write your code the same way you write the synchronous code because the compiler takes care of all the complexity and frees the programmer to write program logic.

```
async Task<int> AccessTheWebAsync()
{
    HttpClient client = new HttpClient();
    Task<string> getStringTask = client.GetStringAsync("http://msdn.microsoft.com");
    DoIndependentWork();
    string urlContents = await getStringTask;
    return urlContents.Length;
}
```

**Figure 9.** Asynchronous routine.

To use this routine, you should expect your return using the await keyword, as in the example: "string urlContents = await client.GetStringAsync ()". Following these guidelines, when the compiler encounters a method with the expression await, which marks the point at which the method cannot continue until the expected asynchronous operation completes, it starts running in the background and continues to run other tasks. When the routine is complete, execution returns on the instruction following the routine call.

C # language enhancements with the async and await keywords restore the sequential order of code while using system resources efficiently. There are still some relevant aspects to be observed, such as concurrency or task synchronization, but these are minor compared to the work required to create a good program that uses parallel processing. Parallelism is to divide for multiple processors that work concurrently; parallelism is used for computing tasks and accelerates the efficiency [103].

The use of the techniques described here helps a lot in the creation of programs that make use of parallel computing and that better use the resources of the system.

*3.5. The Software Developed*

In the context of pension benefits, the actuarial and financial assumptions represent a formal set of estimates for events: biometric, financial, economic, demographic, social, etc. The choice and use of actuarial assumptions that are not committed to the reality to which the participants, sponsors, and entities are subject may lead to incorrect costs, leading to technical deficits or surpluses, as well as overexposure to risks or underexposure to them. The use of more conservative assumptions may lead to higher initial costs, albeit with lower risks of rising costs. On the other hand, the adoption of less conservative assumptions should be made with the knowledge of the risk that they may not be confirmed, allowing for critical solvency problems at a future date.

The actuarial assumptions will always be criteria that are preferably permeated by common sense, remembering that excess safety margins are as burdensome as the excesses of risks that one intends to assume. Both lead to the incapacity to pay, sometimes participants and/or sponsors, or the own provident entity or benefit plan.

Figure 10 summarizes the classification of the actuarial assumptions of the armed forces. The economic and biometric assumptions make up the active choices, given that the manager has an effective capacity to interfere with them, modifying them to each actuarial evaluation, and choosing them according to their perception. Generic premises, on the other hand, do not submit to the manager's perception, varying according to an external reality.

The concepts of parallel computing presented were used in the armed forces actuarial software V 1.1. (AFAS), giving it the ability to demand a low computational effort to produce the results of the actuarial projections. Figure 11 shows the AFAS execution flow for the calculation of the actuarial projections, showing how the data are computed in parallel until the results are obtained.

The data stored in the application DB are put into memory by a parallel data loading process, filling all the data structures required for the calculations. Once these structures are complete, three parallel large tasks (TASK) demand the necessary data from the RAM memory much faster than the HD. The tasks are carried out simultaneously, the ACTIVE MILITARY task being responsible for the calculations referring to the active military, the INACTIVE MILITARY task responsible for the calculations referring to the inactive military, and the PENSIONER task responsible for calculations referring to military pensioners.

Each TASK above performs for each operation on the lists (arrays) in which iterations are executed in parallel and the state of the loop is monitored and manipulated. In addition, the parallel task library (PTL) dynamically scales the degree of concurrency to use all available processors in the most efficient way.

The work developed focused on optimizing the computational application of the actuarial calculation to produce the actuarial projections of the pensioners of the armed

forces. In this sense, any and all access to the hard disk (HD) was eliminated during the execution of the calculations.

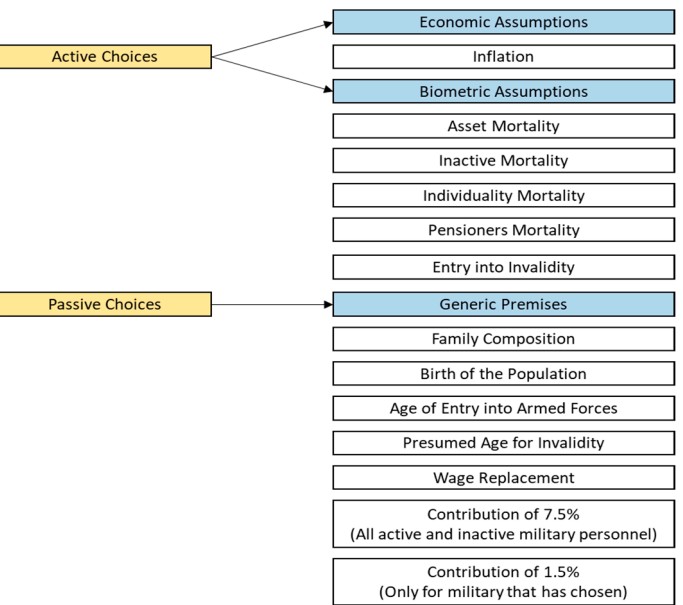

**Figure 10.** Actuarial assumptions of the AF.

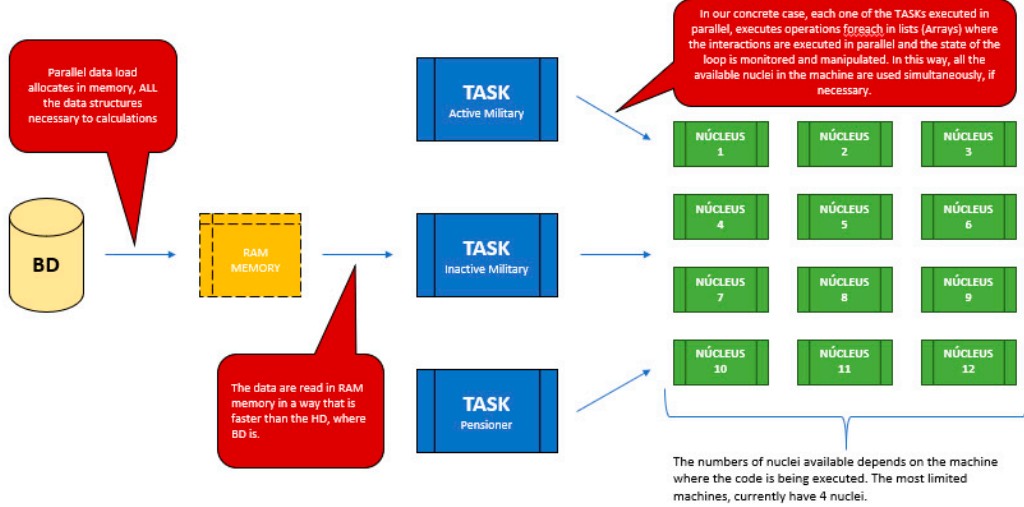

**Figure 11.** Armed forces actuarial software (AFAS).

To achieve this, a technique was developed which loads all the data necessary for the calculations into the computer memory as soon as the application is started. During the application load the data are read from the respective database tables and are generated from SQL queries, and then persisted in structures (arrays) in memory. Once loaded, the data are only read in these structures until the application is closed, which makes the data reading and writing processes very fast.

This technique proved to be efficient, due to the fact that the HD is much slower than RAM. While a DDR4-2400MHZ (1606R) module communicates with the processor at a theoretical speed of 4200 MB/s, the sequential read speed of the current HDs hardly exceeds the 233 MB/s mark. In addition to this, the HD access time, ie., the time required to locate the information and initiate the transfer, is considerably higher than that of the RAM.

While memory is spoken at access times of less than ten nanoseconds, most HDs work with access times greater than ten milliseconds. This causes the HD performance to be much lower when reading small files scattered around the disk (as is the case with virtual

memory). In many situations, the HD gets to the point of not being able to handle more than two or three hundred requests per second. From the foregoing, it can be understood that the loaded memory arrays already allow their data to be accessed at high speeds, compared to the speeds of access to the hard disk. When the data access methods of such structures are developed using parallel programming techniques, the time required to produce the results is considerably reduced.

Figure 12 shows the AFAS parameter window. In it are inserted the parameters that will compose the actuarial projections.

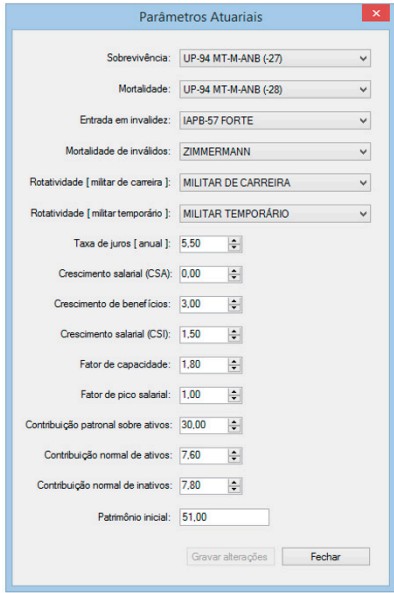

**Figure 12.** AFAS parameters window.

The actuarial projections are important to understand the future behavior of the population and their respective financial flow of payment, using specific actuarial tables for each group and the financial flow.

### 4. Results

For this study, data from the year 2015 for BN, BA, and BAF were used. Thus, more than one and a half million historical records were analyzed regarding the dates of entry into the armed forces, death, disability, birth, and others, whose numbers are subdivided by active and inactive military personnel, pensioners, and beneficiaries. Said amounts are set forth in Table 3.

**Table 3.** Quantitative of military active/inactive and pensioners by 2015.

| Invalidity and Mortality Tables for Invalids Used | | | | | |
|---|---|---|---|---|---|
| IAPB-57 Weak | IAPB-57 Strong | Zimmermann | Zimmermann (Ferr. Germans) | Zimmermann (Empre. Write.) | Grupal Americana |
| Álvaro Comings | TASA-1927 | Prudential (Ferr. Retired.) | IBA (Railways) | Muller | Hunter's |
| IAPB-57 (AJUST/ITAU) | Winklevoss | Bentzien | IAPC | IAPB-57 | ALLG72 |
| USTP61 | Rentiers Français | X17 | - | - | - |

Through the use of the records presented in Table 1 and the actuarial tables matrix, it was possible to develop a C computational application, with the objective of operationalizing the calculations referring to actuarial projections in a fast and accurate manner.

### 4.1. Result of Mortality of Assets, Inactive and Pensioners of the Armed Forces

For the active population, inactive, and pensioners of the three armed forces as a whole, there was no actuarial table that had adherence to the observed mortality data in the database, considering all ages. Therefore, it was necessary to apply the adhesion test to representative samples of this population, with the age range between 20 and 90 years, representing 97% of the population data. Thus, the only table that adhered to the observed mortality, for the ages between 20 and 90 years, was the GKM-70 reduced by 61%, as shown in Figure 13.

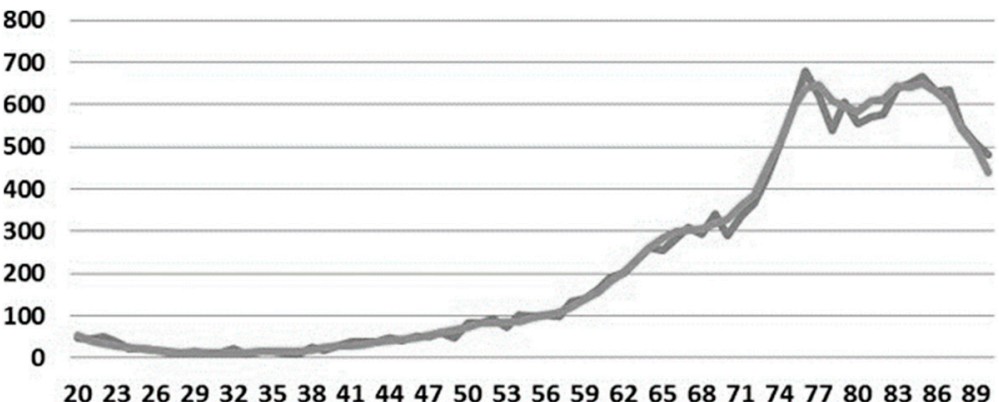

**Figure 13.** Actuarial table for mortality of active, inactive, and pensioners.

### 4.2. Results of Invalidity of Mortality of Armed Forces

For the population of invalids of the armed forces, together, there was no actuarial table that had adherence to observed data of mortality in the database, considering all ages. Therefore, it was necessary to apply the adhesion test to representative samples of this population, with the age range between 21 and 93 years, defined empirically, representing 94% of the population data. Consequently, the tables that adhered to the occurrences of death of invalids were: HUNTER'S for all aggravations between 60% and 76%. The board adopted was HUNTER'S aggravated in 68% because it has the best adherence to the mortality of the disabled of the armed forces, as shown in Figure 14.

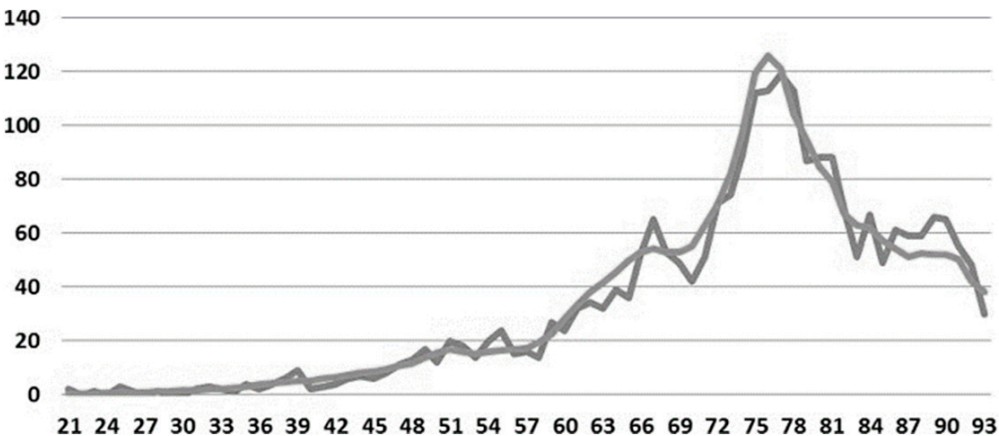

**Figure 14.** Actuarial table for mortality of invalid military personnel.

### 4.3. Result of Entry into Disability of the Armed Forces

The table selected to be used as the input for asset impairment was USTP-61 reduced by 49% because it had the lowest chi-square static. To achieve adherence to this table, it was necessary to analyze exclusively the age range between 25 and 41 years, determined

empirically, otherwise there would be no adherence to any available actuarial table. Such adherence is shown in Figure 15.

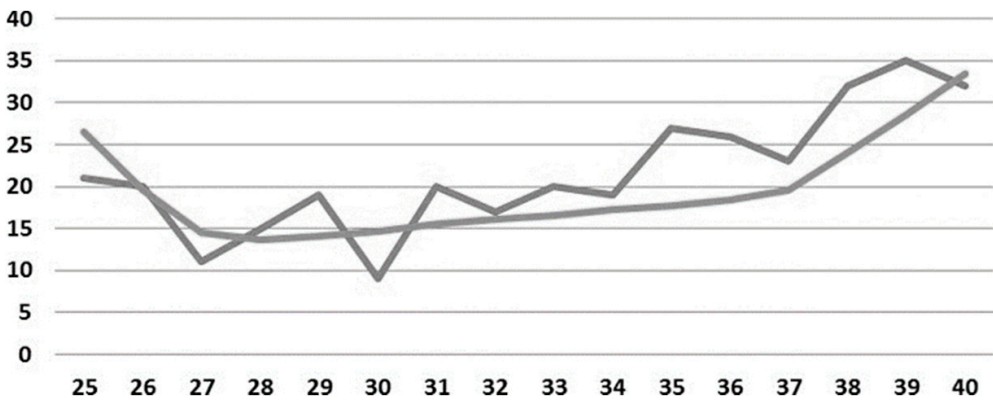

**Figure 15.** Actuarial table for invalid entry of active military personnel.

In addition to the 49% USTP-61 board, which had the best adhesion, the following actuarial tables also accepted the null hypothesis, but with lower accuracy:

- ALLG-72 for all redemptions between 28% and 39%;
- USTP-61 for all redemptions between 38% and 48%; and
- X17 for all redemptions between 50% and 55%.

Figure 16 shows the decrease in the annual financial cost, mainly due to the fact that it is a population that does not consider the entry of new active military personnel; that is, a closed population. Only those related to the payment of pensions, which have an annual financial balance maintained negative, tending to zero, until the population's extinction were considered as costs. This contributes to the validation of the method proposed in this paper.

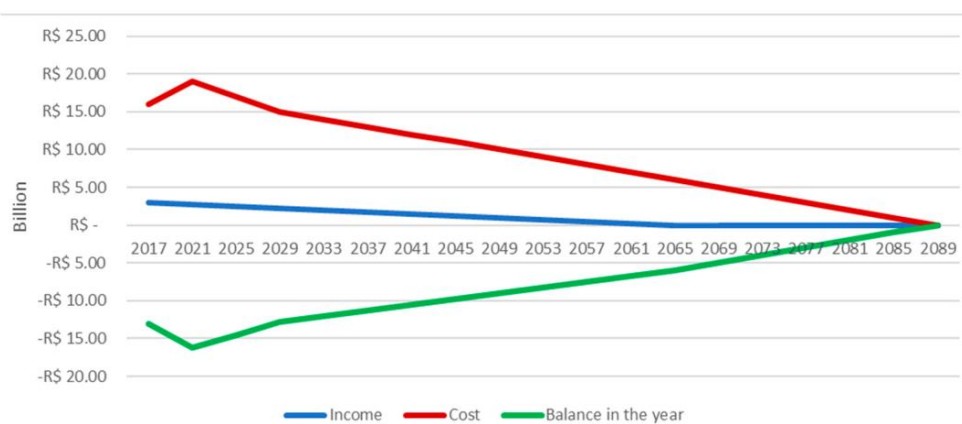

**Figure 16.** Actuarial projection of revenues and costs, without replacement—for pensioners—AF.

*4.4. Processing Time Reduction Results*

Table 4 shows the time required for the production of results by the application of the actuarial calculation in two versions. The first version, "Previous Version", does not contemplate any of the techniques, patterns, or features presented in this chapter. The second version, called "Refactored Version", is the final product of the work presented here. In addition, it shows that in all groups analyzed, there was a significant reduction in the time needed to produce the results of the actuarial calculation, notably in the actuarial calculation of present value. The data are presented in three distinct groups, these groups being notably those that require longer processing time, and whose results are of interest to the study under study.

**Table 4.** Comparison of processing times.

| Process | Previous Version [Approximate] HH:MM:SS | Refactored Version [Approximate] HH:MM:SS | Time Reduction (%) |
|---|---|---|---|
| Importing databases. | 04:00:00 | 00:08:00 | 96.7% |
| Actuarial calculation of present value. | 23:00:00 | 00:07:00 | 99.5% |
| Actuarial projection with a term of 75 years. | 16:00:00 | 00:05:00 | 99.5% |
| Total Time | 43:00:00 | 00:20:00 | 99.2% |

In this context, the use of parallel computing techniques and resources proved to be a solution to reduce the computational effort, allowing the use of all the available processing capacity in the computers. An actuarial projection that used to take 43 h can now be completed in 20 min, which represents a reduction of more than 99%. In the specific case of the Brazilian Army, from a mass of data of 500,000 records, it was possible to execute a complex mathematical model with high recursion in about 5 min.

By way of comparison, the results of reducing processing times obtained in this study are better than those presented in the literature, such as those presented in Section 3.4, which shows the relevance and importance of the created software. The percentages of reduction in processing times of the previous studies are consolidated in Table 5.

**Table 5.** Comparison of processing times with previous studies.

| Papers | Time Reduction (%) |
|---|---|
| Cader et al. [95] | 97.7% |
| Hossain and Assiri [96] | 66.7% |
| López et al. [97] | 96.7 |
| Morishima [98] | 90.9% |
| Zhang [99] | 98.4% |
| Our software (AFAS) | 99.2% |

Analyzing the data in the table above, it is observed that the AFAS presents a superior performance to the systems developed in previous studies, considering the reduction of more than 99% in the processing time. In addition, the created tool allows for finding the actuarial tables that best represent the occurrences of mortality and disability in the military, which makes it unique in the context of the armed forces.

In short, AFAS provides a better dimensioning of military pension costs to be paid by the military pension system in a feasible time due to the application of parallel computing. We emphasize that the implementation of this software allowed the Brazilian Armed Forces to measure the costs of military pensions in a feasible time, which was unfeasible before the creation of the AFAS.

## 5. Conclusions

Through the development and implementation of actuarial software that optimizes the selection of actuarial mortality and disability tables using the concept of multidimensional arrays and programming structures, we were able to determine the actuarial table that provides the greatest precision and accuracy of the decremental events of a population under study. It should be noted that the greater the adherence, the greater the financial efficiency.

Considering the reduction in processing time through the use of parallel computing, the software proposed in this article performed better than those already presented in the literature, presenting a reduction of more than 99%, allowing a feasible time to carry out actuarial forecasts.

We highlight that no technical studies were found on how to optimize the estimation of the actuarial tables to be adopted, considering aggravations and reductions in the probabilities of death and disability, which is widely accepted by current legislation. Therefore, it is considered the unpublished and unique study in the academic community.

We emphasize that the methodology proposed by this research can be applied in the order actuarial problems. As a limitation of this study, we emphasize that the database was restricted only to the Brazilian Armed Forces. Future studies could apply the proposed methodology in other databases of pensioners, in view of the multidisciplinarity of the presented software.

**Author Contributions:** Conceptualization, M.d.S. and C.F.S.G.; methodology, M.d.S. and E.L.P.J.; software, M.Â.L.M. and I.P.d.A.C.; validation, M.Â.L.M. and I.P.d.A.C.; formal analysis, M.Â.L.M. and E.L.P.J. investigation, M.d.S. and L.P.F.; resources, E.L.P.J. and L.P.F.; data curation, L.P.F.; project administration, C.F.S.G. and M.d.S.; funding acquisition, M.d.S., C.F.S.G. and E.L.P.J. All authors have read and agreed to the published version of the manuscript.

**Funding:** This research received no external funding.

**Institutional Review Board Statement:** Not applicable.

**Informed Consent Statement:** Not applicable.

**Data Availability Statement:** Not applicable.

**Conflicts of Interest:** The authors declare no conflict of interest.

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
