# Peer review of "Proposal for Mathematical and Parallel Computing Modeling as a Decision Support System for Actuarial Sciences"

_axioms, doi:10.3390/axioms12030251_

Round 1
Reviewer 1 Report
The subject of the work - Selection of Actuarial Tables for the Brazilian Armed Forces - is interesting and actual. The paper is technically and theoretically sound and well-organized. The structure of the article is transparent. The Metodology and Results are clearly describe. Concluding, the work appears to be worth publishing in presented form.
Author Response
Our specific considerations are attached.

Reviewer 2 Report
Dear Authors
With the usual compliments, I congratulate you on the research initiative that aims to realize a study of the prediction of the financial impacts that the deaths and invalidity of the military of the Brazilian armed forces have on the entire social security system. The theme is relevant and is of practical application to real-life problems. The work is well structured. The methodology is clear. However, I will make some suggestions for improvement:
1) In the abstract and in the conclusion, I suggest making a clear inclusion of the limitations found and future research;
2) Figures 4-6 are not of good quality. I suggest improving them.
3) The graphs represented in figures 8-11 could be better presented if they were arranged in landscape orientation.
4) I suggest that the authors reinforce the argument contained in figures 27-34 with the insertion of the following literature:
(1) https://doi.org/10.1108/DTA-11-2021-0315 (2) https://doi.org/10.1108/JM2-05-2020-0122; (3) https://doi.org/10.1108/IJQRM-04-2021-0114;(4) https://doi.org/10.1590/0103-6513.20190032
5) In the titles of the sections "3.2. Heuristics (Stage 2)," it is relevant to insert the stages of the methodology they represent. I believe it can be removed from all.
Finally,
I wish you a good revision.
Revisor.
Author Response

(The authors gave the same response as above.)

Reviewer 3 Report
Dear Authors,
Starting from the abstract section, you are talking about optimization and the best model implementation, but there is no confirmation that these results have been achieved.
My comments are listed in the attachment.
Kind regards,
Reviewer

Author Response

(The authors gave the same response as above.)

Round 2
Reviewer 2 Report
Dear Authors
Initially, I congratulate you for the improvements implemented and for attending to the reviewers' suggestions. Only in relation to item 4 of the first revision was it not verified due to an initial error. In this sense, I suggest that you consider the insertion of the following references to reinforce the arguments presented in lines 35 to 37 (The decision-making process in political environments involves different areas, interconnecting strategic, tactical, and operational levels in favor of a direction aligned with the objectives in a given problematic situation"):
(1) https://doi.org/10.1108/DTA-11-2021-0315;
(2) https://doi.org/10.1108/JM2-05-2020-0122;
(3) https://doi.org/10.1108/IJQRM-04-2021-0114; and
(4) https://doi.org/10.1590/0103-6513.20190032.
Best regards
Reviewer
Author Response
Our comments are attached.

Reviewer 3 Report
Dear Authors,
You must present more accurately your main study result – the developed Armed Forces Actuarial Software (AFAS), which calculates the actuarial table that provides the highest precision and accuracy of the detrimental events mortality, and disability of the Brazilian military population. Figures 12, 13, and 14 are not suitable to prove your results. In addition, please enhance your results, through numerical comparison to emphasize the superiority of developed AFAS.
This is scientific research and you must present your results clearly. The graphical representation you choose to prove your achievements are not suitable for a high-quality journal. All figures are very low quality. Please enhance the discussion of your results, through numerical comparison to emphasize the superiority of this work, and supplement some prospects and follow-up work.
Conclusions must be short and clear. You must emphasize the importance and uniqueness of the used methodology, which cannot be achieved by using the existing approach.
Sincerely,
Reviewer
Author Response
Our comments are attached.

Round 3
Reviewer 3 Report
Dear Authors,
main study result – the developed Armed Forces Actuarial Software (AFAS), which calculates the actuarial table that provides the highest precision and accuracy of the detrimental events mortality, and disability of the Brazilian military population, is inappropriately presented. Your charts 7, 8, 9, 11, and 12 are not suitable to prove conducted study results. The conducted study results must be enhanced through numerical comparison to emphasize the superiority of developed AFAS.
This is scientific research and you must present your results clearly. The charts in a high-quality journal can't be low quality. Also, it is very important to present your achievements through numerical comparison to emphasize the superiority of this work and supplement some prospects and follow-up work.
The conclusions are shortened, but didn't emphasize the importance and uniqueness of the used methodology.
Sincerely,
Reviewer

Author Response
Our comments are attached.

Round 4
Reviewer 3 Report
Dear Authors,
I checked the revised and resubmitted paper and realized that the improvements were not made. You did not address the suggestions for your paper corrections. Unfortunately, the manuscript is not ready for publication.
Kind regards,
Reviewer
Author Response
Dear Reviewer,
First, I would like to thank you for considering our research for the Axioms journal.
In this way, all figures that were of low quality were modified.
As for the references, these were also changed as indicated by the editorial board.
In this context, I am submitting a new version of the study for your appreciation.
Best Regards